# Wheat and Faba Bean Intercropping Together with Nitrogen Modulation Is a Good Option for Balancing the Trade-Off Relationship between Grain Yield and Quality in the Southwest of China

Ying-an Zhu [1,†], Jianyang He [2,†], Zhongying Yu [2], Dong Zhou [2], Haiye Li [2], Xinyu Wu [2], Yan Dong [2], Li Tang [2], Yi Zheng [2,3] and Jingxiu Xiao [2,*]

1 College of Horticulture and Landscape, Yunnan Agricultural University, Kunming 650201, China
2 College of Resources and Environment, Yunnan Agricultural University, Kunming 650201, China
3 Department of President Office, Yunnan Open University, Kunming 650599, China
* Correspondence: xiaojingxiuxjx@126.com
† These authors contributed equally to this work.

**Abstract:** Cereal and legume intercropping could improve cereal yield, but the role of intercropping in grain quality still lacks a full understanding. A two-year bi-factorial trial was conducted to investigate the role of two planting patterns (mono-cropped wheat (MW) and intercropped wheat+faba bean (IW)) and four nitrogen (N) fertilization levels (N0, no N fertilizer applied to both wheat and faba bean; N1, 90 and 45 kg N ha$^{-1}$ applied to wheat and faba bean; N2, 180 and 90 kg N ha$^{-1}$ applied to wheat and faba bean; N3, 270 and 135 kg N ha$^{-1}$ applied to wheat and faba bean), as well as their interaction on the productivity of wheat grain yield (GY) and quality. The results showed that intercropping increased both the yields of wheat grain protein and amino acids (AAs) relative to MW in both years. No difference in Aas content between IW and MW was found but the 9% grain protein content (GPC) of IW was higher than that of MW in 2020. By contrast, wheat gliadin content was increased by 8–14% when wheat was intercropped with faba bean in both years, and some AAs fractions including essential and non-essential AAs were increased under N0 and N1 levels but declined at the N3 level. This means that intercropping increased the grain quality either for protein and AAs content or for fractions. There was no negative relationship between GPC and GY in the present study, and intercropping tended to increase GPC with increasing GY. In conclusion, wheat and faba bean mainly affected GPC and fractions rather than AAs, and intercropping presented a potential to improve both wheat quality and yield concurrently. Modulated N rates benefitted the stimulation of intercropping advantages in terms of grain yield and quality in the southwest of China and similar regions.

**Keywords:** wheat and faba bean intercropping; grain protein content; protein fractions; profile of amino acids; nitrogen fertilization

## 1. Introduction

Traditional planting patterns including intercropping, relay intercropping, and rotation are normally linked with yield increase and sustainability of the agriculture system [1–3]. Legume-based intercropping, a worldwide planting method, always presents increased crop yield and drives higher crude protein yields due to the nitrogen (N) biological fixation of legumes [4–6]. Frequently, improved cereal nutrient was observed because of N and phosphorus (P) transfer from the legume to cereal during their co-growing period in cereal-legume intercropping systems [7], and resulted in better cereal feed/forage quality [8–10]. Thus, the early research argued that the increased protein content of cereals was a result of N fertilization and was linked with legume intercropping [11,12]. Actually, other non-legume-based intercropping was also a benefit for crops yield and quality [13,14].

Protein content and fractions are important for evaluating and determining wheat grain values [15]. Many researchers highlighted the positive effect of cereal and legume intercropping on grain protein content (GPC) [5,16,17], but few studies focused on the effect of intercropping on protein fractions. The content of amino acids (AAs), especially essential amino acids (EAAs), is important to reflect protein quality, but major staple foods including wheat have limited amounts of EAAs for humans [18]. The enhancement of breeding techniques and N topdressing time modulation resulted in improved protein quality [19,20]; however, little attention has focused on the role of planting pattern in grain AAs content and factions.

Wheat and faba bean intercropping, as a typical legume-based intercropping pattern, is widely distributed in many countries either for food or for forage [21]. Tosti and Guiducci observed that wheat temporarily intercroped with faba bean improved both wheat grain yield and protein content [22]. However, De Stefanis et al. found that durum wheat gluten quality, total protein concentration, and monomeric and polymeric protein amounts were significantly increased but wheat grain yield was decreased when durum wheat was temporary intercropped with faba bean [23]. Similarly, wheat temporarily intercropped with clover induced a higher wheat grain protein content but lower grain yield [24]. In fact, a negative relationship or trade-off relationship between the grain yield (GY) and GPC was constantly observed in most cereal grains [25,26], but intercropping was a good strategy to reducing the risk of impairing winter wheat yield and protein content [27].

In the southwest of China, wheat and faba bean had a long co-growing period; thus, the interspecific interaction in this pattern was different from that of wheat temporarily intercropped with faba bean [21]. A previous study illustrated that wheat and faba bean intercropping could increase wheat yield but decrease faba bean yield, and the intercropping yield advantage was decreased with N input [28]. However, there is a lack of comprehensive assessment on the effect of intercropping on grain quality, especially on the content and fractions of wheat protein and AAs, which are tightly related to N input. We hypothesize that wheat and faba bean intercropping could improve wheat grain yield and maintain gain quality simultaneously, and the effect of intercropping on grain quality would vary with N input. Here, we present a two-year field experiment to test the hypothesis: (i) qualifying the effect of intercropping on wheat grain protein and amino acids under different N input conditions, and (ii) identifying the impact of intercropping on the relationship between GY and quality.

## 2. Material and Methods

### 2.1. Experimental Site and Growing Conditions

The present study was based on the data collected during 2018/2019 and 2019/2020 cropping seasons in the existing wheat and faba bean intercropping experiment, which was established in 2014. The field experiment was conducted at the Yunnan Agricultural University research station, located in Xundian (23°32′ N, 103°13′ E), Yunnan Province, northwest China. The climate in this region is characterized by a unimodal rainfall pattern with a rainy season from June to September and mean annual rainfall of 1040 mm, and the mean annual air temperature is 14.7 °C. The average monthly temperatures and monthly precipitation amounts during the experiment of 2018/2019 and 2019/2020 are shown in Figure 1. The monoculture corn was planted from May to September for many years before the wheat and faba bean intercropping experiment was established. The soil type in this region is called red soil (Ferralic Cambisol, FAO, 2006) with a bulk density of 1.38 g cm$^{-3}$, and the content of clay, silt, and sand was 34%, 52%, and 14%, respectively, at a soil depth of 0–30 cm. At the beginning of the multi-year field experiment in 2014, the soil properties were as follows: SOC 12 g kg$^{-1}$, total N 1.14g kg$^{-1}$, total P 0.98g kg$^{-1}$, total K 24.25 g kg$^{-1}$, available N (NaOH hydrolyzed) 80 mg kg$^{-1}$, Olsen P 17 mg kg$^{-1}$, exchangeable K 146 mg kg$^{-1}$, and pH 7.2 (1:2.5 soil: water). The soil total N and available N contents in each treatment were changed during 2018/2019 and 2019/2020 cropping seasons as compared to the beginning of the field experiment in 2014 due to continuous wheat and faba bean intercropping and different N application rates (data shown in Supplementary Table).

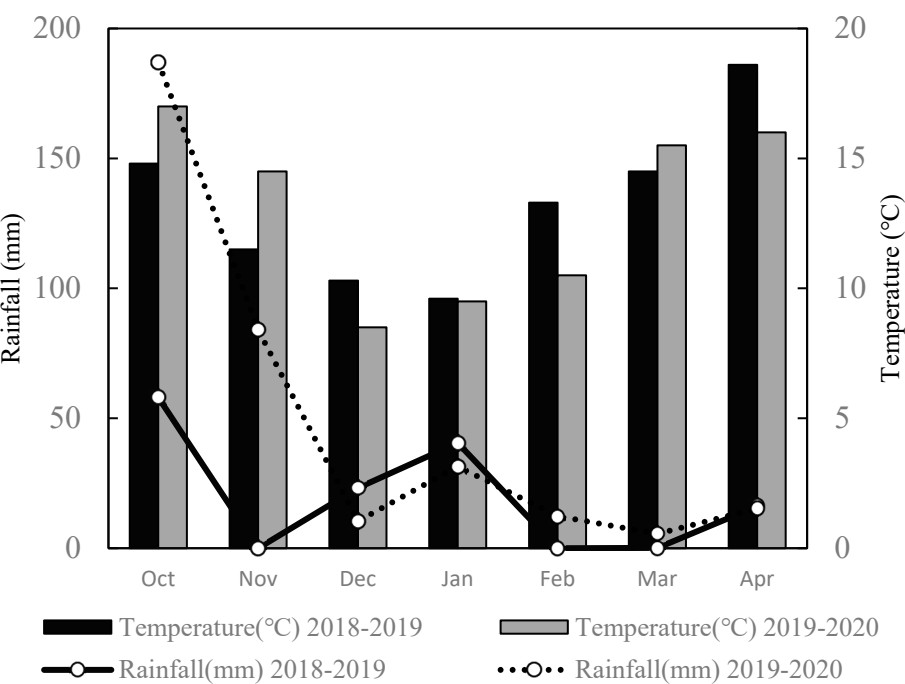

**Figure 1.** The monthly average temperature and rainfall during experiments of 2018–2019 and 2019–2020.

## 2.2. Experimental Design

The field experiment was a randomized block design with two factors and three replicates [28]. Factor A was planting patterns (mono-cropped wheat (MW) and intercropped wheat+faba bean (IW)), and factor B was N levels (0 kg N ha$^{-1}$ (N0), 90 kg N ha$^{-1}$(N1), 180 kg N ha$^{-1}$ (N2), and 270 kg N ha$^{-1}$ (N3) for wheat; 0 kg N ha$^{-1}$ (N0), 45 kg N ha$^{-1}$(N1), 90 kg N ha$^{-1}$(N2), and 135 kg N ha$^{-1}$ (N3) for faba bean). In total, the field experiments consisted of 24 plots with eight treatments, and each plot area was 5.4 m × 6.0 m = 32.4 m$^2$. There were 0.5 m spacings between each plot and 1.0 m spacings between adjacent blocks to avoid water and nutrient interference. The row space of wheat was 0.2 m with a seeding rate at 180 kg ha$^{-1}$, whereas the faba bean row spacing was 0.3 m and the plant-to-plant spacing was 0.1 m in the present study. The strip intercropping of six rows of wheat intercropped with two rows of faba bean was used in this study based on local farmers practice; thus, there were three strips in each intercropping plot including 18 rows of wheat and six rows of faba bean [28]. The plant density of intercropped wheat and faba bean was identical to that of mono-cropped under the same area, and the row space between wheat and faba bean was 0.25 m in each intercropped plot. Detailed information of a given intercropping plot can be seen in Figure 2.

## 2.3. Field Experiment Management

The local varieties of Yunmai 52 for wheat (*Triticum aestivum* L.) and Yuxi Dalidou for faba bean (*Vicia faba* L.) were used in the present study since 2014, and the faba bean seed was non-inoculated rhizobium. Wheat and faba bean were sown on the same date normally on 20–30 October with a sowing depth of 10 cm and were harvested in the next year on 10–20 April. After both plants were harvested, all straws were removed from the field and each plot retained fallow from May to September since 2014. The implementation of other crop managements including irrigation and the use of pesticides was according to local farmers' practice.

Urea as N fertilizer was used in the present study. For wheat, one half of the total N application rate for each given treatment was applied as basal fertilizer before sowing by hand, and another half N fertilizer as a topdressing was applied at the wheat elongation stage. For faba bean, all N fertilizers for each treatment were applied as a basal fertilizer before sowing. Amounts of 90 kg P$_2$O$_5$ ha$^{-1}$ (calcium superphosphate) and 90 kg K$_2$O ha$^{-1}$

(potassium chloride) for each crop were applied as base fertilizers according to local farming practices. In each intercropping plot, topdressing N was only evenly applied to wheat rows by hand.

### 2.4. Data Collection and Analyses

At maturity, inter- and mono-cropped wheat grains of each whole plot were collected and determined after the grain seeds were fully air-dried during 2018/2019 and 2019/2020 growing seasons, and the experiment of 2019 and 2020 represented two years of experiments, respectively. The wheat grain crude protein; protein fraction contents including albumin, globulin, gliadin, and glutelin; amino acids fraction content were determined in both years.

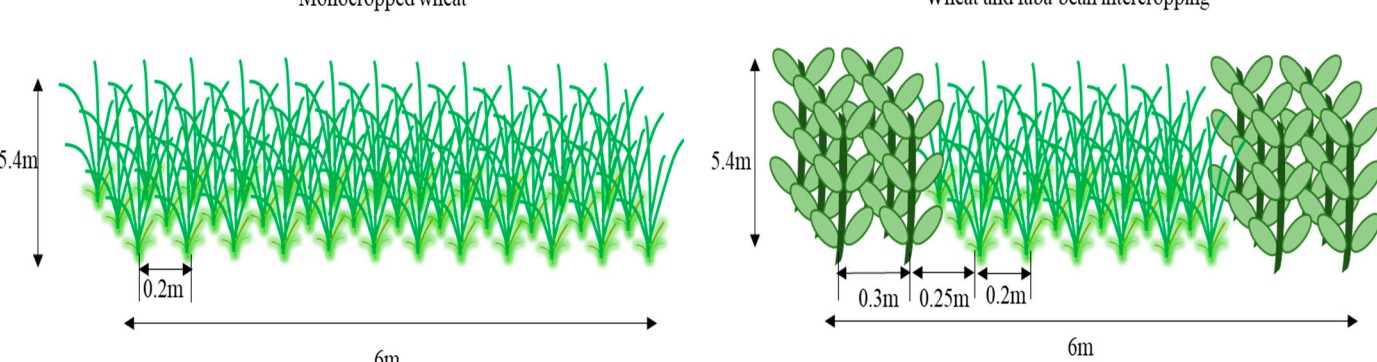

**Figure 2.** Diagram of the planting pattern for wheat and faba bean intercropping and mono-cropped wheat in the field experiment.

GPC was calculated by multiplying the grain N content with a conversion factor of 5.83 for wheat [29]. Grain N content was analyzed by the Kjeldahl method after the sample digestion with $H_2SO_4$-$H_2O_2$. Protein fractions albumin, globulin, gliadin, and glutelin were sequentially extracted from 1 g of wheat grain powder [30,31]. In brief, sequential extraction of albumin and gliadin fractions from the wheat grain sample were carried out by using distilled water and 2% NaCl, followed by extraction with 70% ethanol to obtain the gliadin fraction. The glutelin fraction was extracted from the residue by using 0.05 M NaOH. Protein content was determined using the modified Lowry method of Markwell et al. [32].

Amino acids (AAs) were identified and quantified by a high-performance liquid chromatographer (Agilent 1100) coupled to a DAD detector and a post-column derivatization device. The chromatograph column used was a C18 (250 × 4.6 mm ID) from Thermo Fisher, and the column was operated at a temperature of 40 °C. The chromatograph conditions were set as follows: ultraviolet detector 360 nm; flowrate 1.0 mL min$^{-1}$; the mobile phase consisted of A = 0.5 M sodium acetate (for HPLC analysis, Sigma Chemical CO., St. Louis, MO, USA) and B = 50% (*v/v*) methanol (Sigma Chemical Co., St. Louis, MO, USA) in water, and the injection volume was 10 μL for all samples. Wheat grain powder was hydrolytic and derivatized before HPLC analysis. Briefly, (1) sample hydrolysis: grain powder was hydrolytic for 24 h at 110 °C in 6M hydrochloric acid; (2) post-column derivatization: the derivatization was performed from a solution containing sodium hydroxide (6 mol L$^{-1}$), sodium bicarbonate pH 9.0 (0.5 mol L$^{-1}$), and DNFB. A deviation solution was mixed in a buffer of phosphoric acid pH 7.0 and filtered with a 0.22 μm membrane before HPLC analysis. The identification of amino acids was carried out by comparing retention times of standards and quantification in analytical curves constructed for each amino acid.

The sum content of seventeen AAs was the total AAs (TAAs) content. The seventeen measured AAs were divided into essential amino acids (EAAs) and non-essential amino acid (NEAAs). EAAs are essential for humans and animals but cannot be synthesized in the human body, including Thr, Val, Met, Ile, Leu, Phe, and Lys; NEAAs are non-essential

as being synthesized in the human or animal body, including Asp, Ser, Glu, Gly, Ala, Cys, Tyr, His, Arg, and Pro [33].

Protein and TAAs yields represent the yield of protein and/or TAA that can be harvested per unit area of crops [34], which was calculated by protein and AAs content multiplied by each plot grain yield, respectively, in this study.

### 2.5. Statistic Analysis

A two-way analysis of variance (ANOVA) was performed using the MIXED procedure with SPSS software (IBM SPSS Statistics Version 19.0) to test for significant differences among treatments. Planting patterns and N levels were considered as the fixed factors, and replication was considered the random factor. Significant differences among treatments at each year were investigated using Duncan's multiple range post hoc test when the F-value was significant ($p \leq 0.05$). Linear and quadratic models were used to simulate the relationship among grain yield, grain protein content, and grain AAs content in this study.

## 3. Results

### 3.1. Mono- and Inter-Cropped Wheat Grain Protein Content and Yield under Different N Levels

Both the GPC and protein yield were not influenced by the interaction of N levels and planting patterns in the two-year field experiments. Likewise, N levels and planting patterns also had no impact on wheat GPC in the experiment of 2019. However, wheat GPC was increased by 9% when wheat was intercropped with faba bean relative to MW in the experiment of 2020 (Table 1). Similarly, wheat protein yield was increased by 28% and 32% in 2019 and 2020, respectively, when wheat was intercropped with faba bean. In addition, increased protein yield was found with increasing N levels in both years (Table 1).

**Table 1.** The protein and total amino acids content and yield for inter- and mono-cropped wheat grain under different N levels.

| N Levels | Planting Patterns | 2019 | | | 2020 | | | 2019 | | 2020 | |
|---|---|---|---|---|---|---|---|---|---|---|---|
| (NL) | (PP) | GY | Protein Content | Protein Yield | GY | Protein Content | Protein Yield | TAAs Content | TAAs Yield | TAAs Content | TAAs Yield |
| | | t ha$^{-1}$ | % | g m$^{-2}$ | t ha$^{-1}$ | % | g m$^{-2}$ | mg g$^{-1}$ | g m$^{-2}$ | mg g$^{-1}$ | g m$^{-2}$ |
| N0 | | 1.69 d | 13 a | 2.17 d | 1.90 c | 10 c | 1.97 d | 92 c | 1.55 d | 81 d | 1.56 d |
| N1 | | 3.08 c | 13 a | 4.11 c | 3.24 b | 10 c | 3.41 c | 99 b | 3.10 c | 86 c | 2.78 c |
| N2 | | 4.02 b | 14 a | 5.45 b | 3.72 a | 12 b | 4.53 b | 95 bc | 3.81 b | 103 b | 3.85 b |
| N3 | | 4.62 a | 13 a | 6.19 a | 3.92 a | 14 a | 5.37 a | 117 a | 5.41 a | 113 a | 4.41 a |
| | MW | 3.08 b | 13 a | 3.92 b | 2.86 b | 11 b | 3.30 b | 100 a | 3.20 b | 96 a | 2.88 b |
| | IW | 3.63 a | 14 a | 5.04 a | 3.53 a | 12 a | 4.34 a | 102 a | 3.74 a | 95 a | 3.42 a |
| N0 | MW | 1.41 a | 13 a | 1.80 a | 1.41 a | 10 a | 1.43 a | 93 c | 1.32 e | 78 e | 1.11 e |
| | IW | 1.98 a | 13 a | 2.54 a | 2.39 a | 11 a | 2.50 a | 91 cd | 1.79 d | 84 d | 2.01 d |
| N1 | MW | 2.67 a | 13 a | 3.44 a | 2.87 a | 9 a | 2.70 a | 85 d | 2.26 c | 85 d | 2.45 c |
| | IW | 3.49 a | 14 a | 4.78 a | 3.63 a | 11 a | 4.12 a | 113 b | 3.95 b | 86 d | 3.11 b |
| N2 | MW | 3.72 a | 13 a | 4.94 a | 3.43 a | 11 a | 3.90 a | 98 bc | 3.66 b | 100 c | 3.42 b |
| | IW | 4.32 a | 14 a | 5.97 a | 4.01 a | 13 a | 5.15 a | 92 cd | 3.96 b | 107 b | 4.29 a |
| N3 | MW | 4.50 a | 12 a | 5.51 a | 3.74 a | 14 a | 5.17 a | 124 a | 5.56 a | 122 a | 4.56 a |
| | IW | 4.74 a | 14 a | 6.87 a | 4.09 a | 14 a | 5.56 a | 111 b | 5.26 a | 105 bc | 4.27 a |
| Sig | | | | | | | | | | | |
| NL | | *** | ns | *** | *** | *** | *** | *** | *** | *** | *** |
| PP | | *** | ns | *** | *** | * | *** | ns | *** | ns | *** |
| NL × PP | | ns | ns | ns | ns | ns | ns | *** | *** | *** | *** |

MW, mono-cropped wheat; IW, inter-cropped wheat; GY, grain yield; TAA, total amino acid. In each column, different letters represent significant differences among treatments at the 0.05 level according to Duncan's multiple range test. * and *** represent significant differences at 0.05 and 0.001 levels, respectively. ns represents no significant difference.

### 3.2. Mono- and Inter-Cropped Wheat Grain Protein Composition under Different N Levels

Four protein fraction contents including albumin, globulin, gliadin, and glutelin were influenced by N levels, and protein fraction contents were frequently affected by the planting pattern, but they were not influenced by the interaction of N levels and planting patterns (Table 2). In 2019, the increased contents of albumin, gliadin, and glutelin in IW grain were observed as compared to MW, and the increase was 9%, 9%, and 5%, respectively.

In 2020, only the increased content of gliadin in IW grain was observed relative to MW, and the increase was 14%. In addition, all four protein fractions were increased with increasing N levels (Figure 3).

**Table 2.** Two-way ANOVA analysis of grain protein composition for inter- and mono-cropped wheat under different N levels.

| | 2019 | | | | 2020 | | | |
|---|---|---|---|---|---|---|---|---|
| | Albumin | Globulin | Gliadin | Glutelin | Albumin | Globulin | Gliadin | Glutelin |
| N levels (NL) | ** | * | *** | *** | *** | *** | *** | *** |
| Planting patterns(PP) | * | ns | ** | * | ns | ns | *** | ns |
| NL×PP | ns | ns | ns | Ns | ns | ns | ** | ns |

In each column, *, **, and *** represent significant differences at 0.05, 0.01, and 0.001 levels, respectively. ns represents no significant difference.

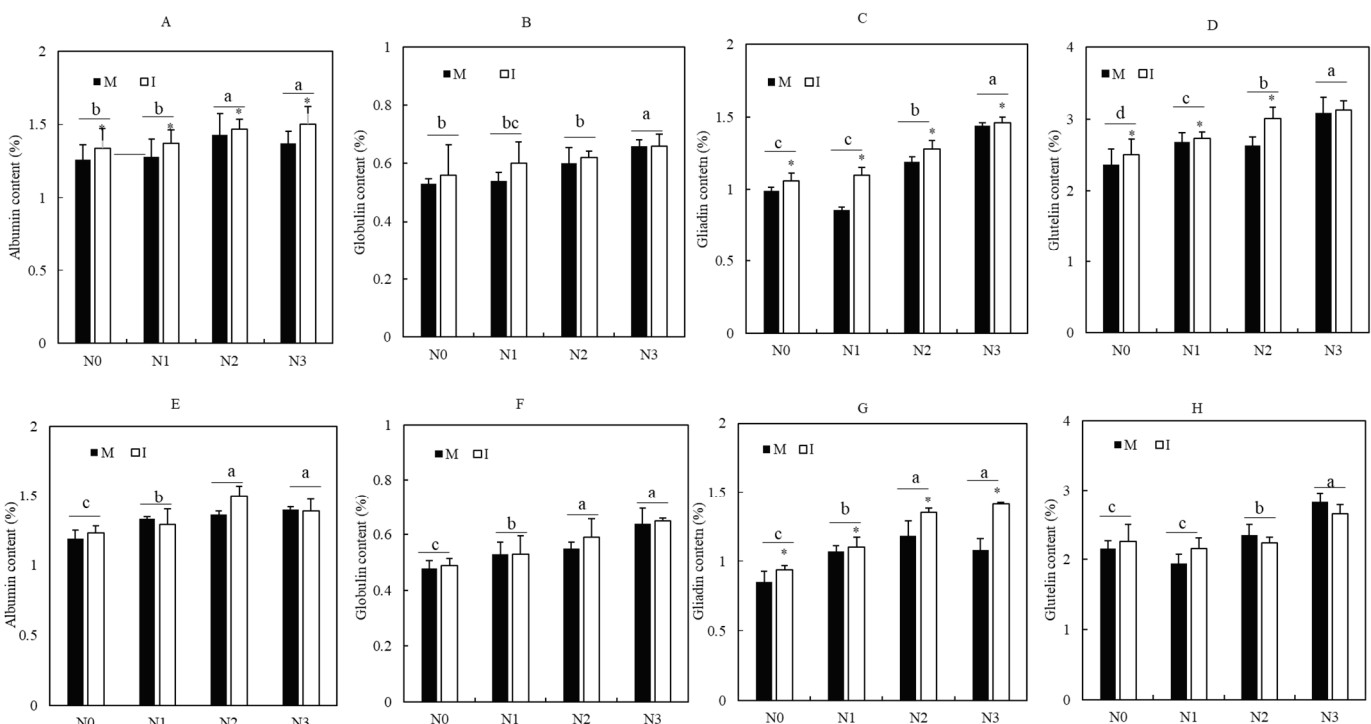

**Figure 3.** Grain protein fraction content between IW and MW under different N levels. MW, mono-cropped wheat; IW, intercropping wheat. (**A–D**) Albumin, globulin, gliadin, and glutelin content of IW and MW in 2019, respectively; (**E–H**) albumin, globulin, gliadin, and glutelin content of IW and MW in 2020, respectively. Different letters represent significant differences among different N levels ($p < 0.05$); * represents significant differences between IW and MW ($p < 0.05$). Each bar in the figures is the mean value (n = 3), and error bars represent the standard error.

*3.3. Mono- and Inter-Cropped Wheat Grain Amino Acids Content and Yield under Different N Levels*

Planting patterns had no impact on grain TAAs content in neither year, but TAAs content was influenced by N levels and the interaction of N levels and planting patterns (Table 1). The TAAs content in IW grain was increased by 33% relative to MW at the N1 level in 2019, and wheat grain TAAs content was increased by 7% under N0 and N1 levels when wheat was intercropped with faba bean as compared to MW in 2020. However, wheat grain TAAs content was decreased by 10% and 14% under the N3 level in the experiment of 2019 and 2020, respectively, as compared to MW. Regardless of N levels, the grain TAAs yield was increased by 17–19% when wheat was intercropped with faba bean, whereas no difference in grain TAAs yield between IW and MW was found under the N3 level, due to

the interaction between N levels and planting patterns. By contrast, the TAAs yield of IW grain was increased by 35% and 60% under N0 and N1 levels, respectively, in comparison with MW in 2019; the TAAs yield of IW grain was increased by 80%, 27%, and 25% under N0, N1, and N2 levels, respectively, in comparison with MW in 2020 (Table 1).

*3.4. Mono- and Inter-Cropped Wheat Grain NEAAs and EAAs Content under Different N Levels*

The content of NEAAs and EAAs and the ratio of EAAs and TAAs were not influenced by planting patterns but were affected by N levels and N levels × planting patterns in both years (Table 3). When compared to MW, IW NEAAs content was decreased by 12% and 14% under the N3 level in 2019 and 2020, respectively (Figure 4). By contrast, the NEAAs content of IW was 31% higher than that of MW under the N1 level in 2019; the NEAAs contents of IW were 7% and 5% higher than those of MW under N0 and N2 levels, respectively, in 2020 (Figure 4). Similarly, the IW EAAs content was decreased by 14% at the N3 level in 2020 and was decreased by 9% and 12% at N0 and N2 levels in 2019 as compared to the corresponding MW, respectively. However, grain EAAs was increased by 39% at the N1 level in 2019 and increased by 13% at the N2 level in 2020 when wheat was intercropped with faba bean. As a result, EAAs/TAAs of IW at N0 and N2 levels were decreased by 7% and 6%, respectively, and the EAAs/TAAs of IW at the N3 level was increased by 5% when compared to MW in 2019. In all, we did not find any difference in EAAs/TAAs between MW and IW regardless of N levels (Figure 4).

**Table 3.** Two-way ANOVA analysis of non-essential amino acids and essential amino acids for inter- and mono-cropped wheat under different N levels.

|  | 2019 | | | 2020 | | |
|---|---|---|---|---|---|---|
|  | **NEAAs** | **EAAs** | **EAAs/TAAs** | **NEAAs** | **EAAs** | **EAAs/TAAs** |
| N levels (NL) | *** | *** | ** | *** | *** | *** |
| Planting patterns (PP) | ns | ns | ns | ns | ns | ns |
| NL × PP | *** | *** | ** | *** | *** | *** |

In each column, ** and *** represent significant differences at 0.01 and 0.001 levels, respectively. ns represents no significant difference.

*3.5. Mono- and Inter-Cropped Wheat Grain AAs Fraction Content under Different N Levels*

The AAs fraction contents including eight EAAs fractions and nine NEAAs fractions were detected in the present study, and they were seldom influenced by planting patterns but were frequently affected by N levels and N levels × planting patterns according to the two-year experiment (Tables 4 and 5). Under N0 and N1 levels, only Met (2019) and Val (2020) contents in IW grain were lower than those in MW; for the other EAAs fractions, wheat and faba bean intercropping either had no impact on EAAs contents or increased EAAs contents. By contrast, under the N3 level, half of the EAAs fraction contents in IW grain were decreased as compared to MW. In the experiment of 2019, Thr, Val, Phe, and Lys contents in the IW grain were decreased by 12%, 40%, 7%, and 9% relative to MW; Val, Met, His, and Lys were decreased by 31%, 28%, 13%, and 26% when compared to MW in the experiment of 2020. On average, the contents of His and Phe in IW grain were higher than those in MW, but the Lys content in IW grain was lower than that in MW in 2019 regardless of N levels. Similarly, no difference in fraction content of EAAs between IW and MW was found except for the His content in IW grain that was higher than that in MW in 2020 (Table 4).

Wheat and faba bean intercropping nearly had no impact on NEAAs fraction contents except for Asp, Pro, Glu, and Tyr in the two-year experiments. Only Asp, Arg, and Cys contents in IW grain at the N0 level and Cys content at the N1 level in 2019 decreased as compared to MW, and the other NEAAs fraction contents in IW grain were either equal to or higher than those in MW. Likewise, only decreased Asp and Ala contents in IW grain in 2019 and decreased Cys in IW grain in 2020 were observed as compared to MW at the

N2 level. By contrast, half of the NEAAs fraction contents in IW grain were decreased in comparison with MW at the N3 level. In all, wheat grain Asp content in 2019 and Gln content in 2020 were decreased when wheat was intercropped with faba bean regardless of N levels, and a similar or higher content for other NEAAs fractions in IW grain was observed relative to MW (Table 5).

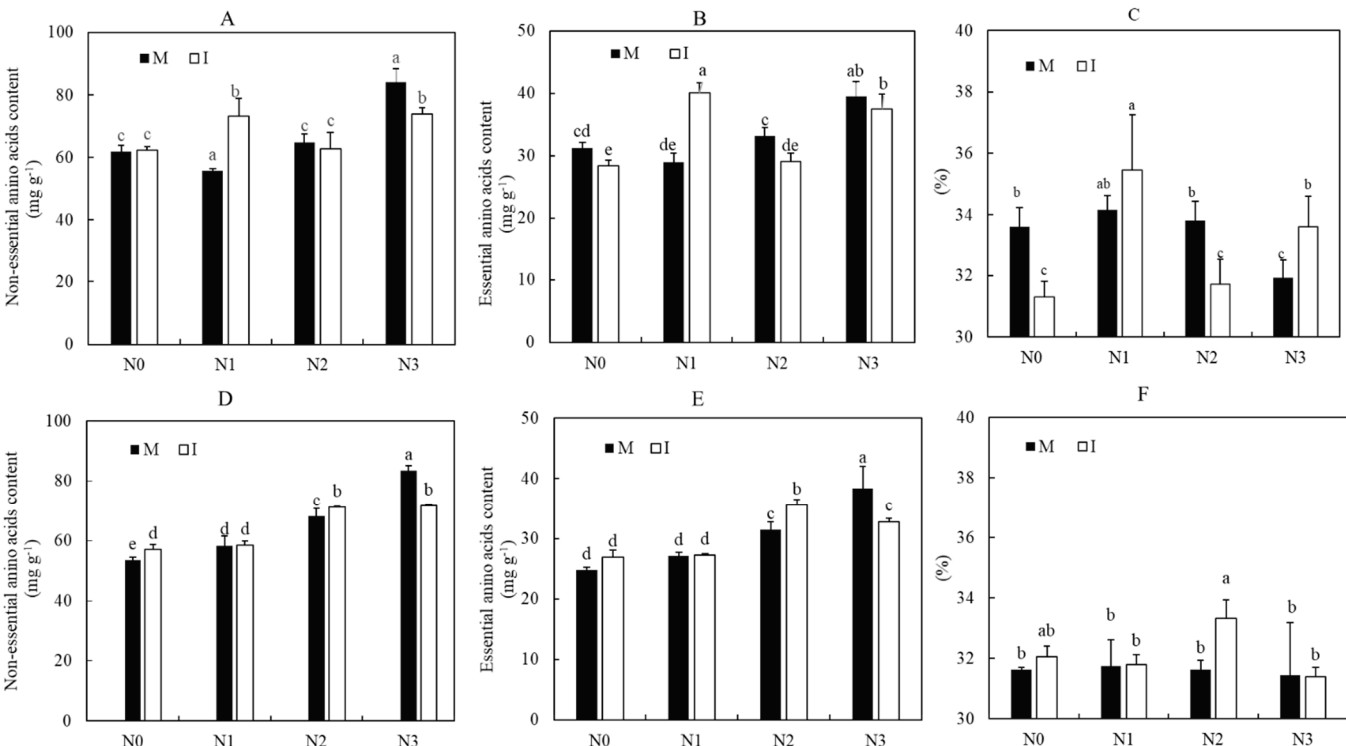

**Figure 4.** Essential amino acids, non-essential amino acids, and the ratio of essential amino acids to total amino acids between IW and MW under different N levels. (**A,D**) Essential amino acids in 2019 and 2020, respectively; (**B,E**) non-essential amino acids in 2019 and 2020, respectively; (**C,F**) ratio of essential amino acids to total amino acids in 2019 and 2020, respectively. MW, mono-cropped wheat; IW, intercropping wheat; different letters represent significant differences among all treatments. Each bar in the figures is the mean value (*n* = 3), and error bars represent standard error.

*3.6. Co-Relationship of Between Grain Yield, Grain Protein Content, and Amino Acids Content for Mono- and Inter-Cropped Wheat*

No relationship between GY and GPC was found for MW, but a quadratic regression was fitted to the relationship between GY and GPC in IW. A positive relationship between GY and AAs content including TAAs, NEAAs, and EAAs was presented, and the AAs content was positively related to GPC (Figure 5).

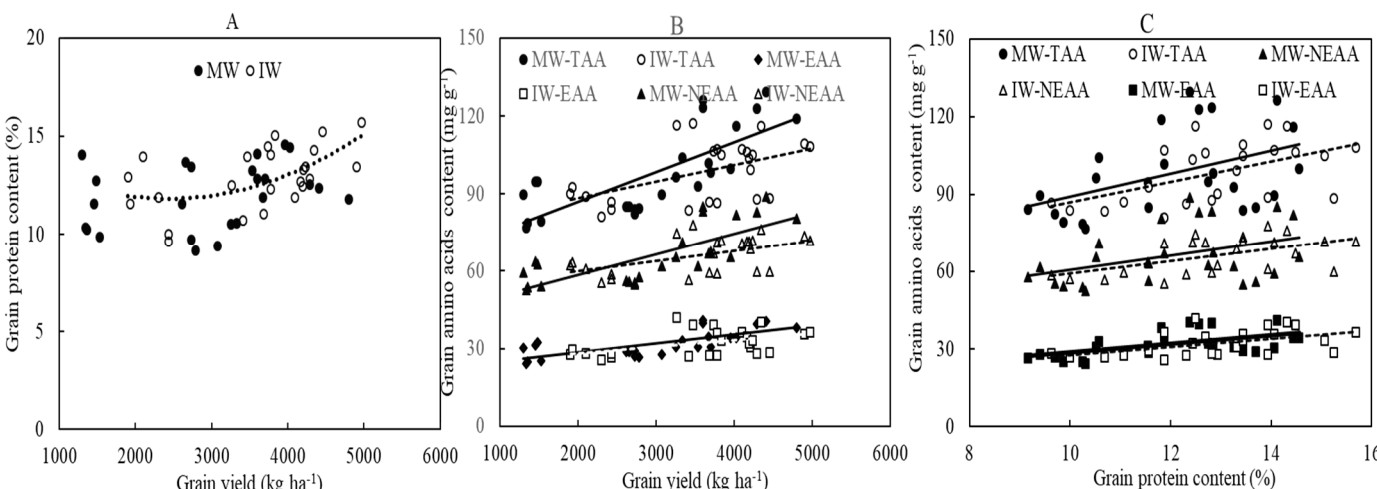

**Figure 5.** Relationship analysis among grain yield, grain protein content, and grain amino acids fraction. (**A**) Relationship between grain yields and grain protein content. In panel A, grain yield as a function of grain protein content for IW (y = 5 × 10$^{-7}$x$^2$ − 0.0025x + 14.833, R$^2$ = 0.3228, *p* = 0.017, *n* = 24); (**B**) relationship between grain yields and grain amino acids fraction. In panel B, grain yield as a function of MW-TAA (y = 0.0115x + 63.997, R$^2$ = 0.5679, *p* = 0.00, *n* = 24), IW−TAA (y = 0.0065x + 75.28, R$^2$ = 0.2636, *p* = 0.01, *n* = 24), MW-NEAA (y = 0.0081x + 42.344, R$^2$ = 0.5676, *p* =0.000, *n* = 24), IW-NEAA (y = 0.004x + 51.937, R$^2$ = 0.2774, *p* = 0.008, *n* = 24), MW-EAA (y = 0.0034x + 21.653, R$^2$ = 0.5219, *p* = 0.00, *n* = 24), and IW-EAA (y = 0.0025x + 23.343, R$^2$ = 0.2135, *p* = 0.023, *n* = 24). (**C**) Relationship between grain protein content and grain amino acids fraction. In panel C, grain yield as a function of MW-TAA (y = 4.435x + 44.956, R$^2$ = 0.2079, *p* = 0.025, *n* = 24), IW-TAA (y = 3.9529x + 47.562, R$^2$ = 0.2924, *p* = 0.006, *n* = 24), MW-NEAA (y = 2.7833x + 32.919, R$^2$ = 0.1662, *p* = 0.048, *n* = 24), IW-NEAA (y = 2.4028x + 35.344, R$^2$ = 0.2968, *p* = 0.006, *n* = 24), MW-EAA (y = 1.6517x + 12.038, R$^2$ = 0.2976, *p* = 0.006, *n* = 24), and IW-EAA (y = 1.5501x + 12.218, R$^2$ = 0.2506, *p* = 0.013, *n* = 24). MW, mono-cropped wheat; IW, intercropping wheat. TAA, NEAA, and EAA: total amino acids, non-essential amino acids, and essential amino acids, respectively. The dot-dashed line and solid line represent linear regressions for IW and MW, respectively.

**Table 4.** The fraction content of each essential amino acids in grain for inter- and mono-cropped wheat grain under different N levels.

| N Levels (NL) | Planting Patterns (PP) | 2019 | | | | | | | | 2020 | | | | | | | |
|---|---|---|---|---|---|---|---|---|---|---|---|---|---|---|---|---|---|
| | | Thr | Val | Met | Ile | Leu | Phe | His | Lys | Thr | Val | Met | Ile | Leu | Phe | His | Lys |
| | | % | | | | | | | | | | | | | | | |
| N0 | MW | 3.45 d | 0.94 b | 7.98 c | 3.45 d | 6.13 d | 4.43 e | 2.20 c | 2.70 a | 2.86 a | 0.16 e | 5.56 c | 3.34 a | 5.39 a | 4.58 a | 1.37 d | 1.50 c |
| | IW | 3.51 cd | 0.80 bc | 5.59 e | 3.54 d | 6.26 d | 4.68 e | 2.04 c | 1.96 c | 3.21 a | 0.50 cd | 5.72 c | 3.36 a | 5.97 a | 4.29 a | 1.92 c | 1.94 b |
| N1 | MW | 3.44 d | 0.65 cd | 6.70 d | 3.45 d | 6.14 d | 4.38 e | 1.84 d | 2.33 b | 3.23 a | 0.53 cd | 6.21 bc | 3.26 a | 5.91 a | 4.46 a | 1.53 d | 1.96 b |
| | IW | 4.38 a | 1.24 a | 10.80 a | 4.20 abc | 8.40 a | 6.00 c | 2.43 b | 2.68 a | 3.34 a | 0.38 d | 5.48 c | 3.44 a | 6.06 a | 4.81 a | 1.88 c | 1.84 b |
| N2 | MW | 3.86 bc | 0.65 cd | 7.98 c | 3.88 bc | 7.23 bc | 5.39 d | 1.73 d | 2.46 ab | 3.63 a | 0.72 b | 7.20 b | 3.57 a | 7.05 a | 5.13 a | 2.33 b | 1.85 b |
| | IW | 3.69 cde | 0.49 e | 5.70 e | 3.70 cd | 6.66 cd | 5.16 d | 1.85 d | 1.84 c | 3.69 a | 0.94 a | 9.67 a | 3.83 a | 7.25 a | 5.13 a | 2.55 a | 2.52 a |
| N3 | MW | 4.51 a | 0.92 b | 9.05 b | 4.70 a | 8.47 a | 7.20 a | 2.45 b | 2.17 b | 4.20 a | 0.96 a | 9.39 a | 4.58 a | 7.78 a | 6.24 a | 2.60 a | 2.65 a |
| | IW | 3.98 b | 0.55 de | 9.21 b | 4.33 ab | 7.99 ab | 6.70 b | 2.69 a | 1.99 c | 3.95 a | 0.66 bc | 6.74 b | 4.14 a | 7.26 a | 5.88 a | 2.25 b | 1.94 b |
| N0 | | 3.48 c | 0.87 a | 6.79 b | 3.49 b | 6.20 c | 4.55 c | 2.12 b | 2.33 b | 3.04 c | 0.33 c | 5.64 b | 3.35 b | 5.68 b | 4.43 c | 1.65 b | 1.72 c |
| N1 | | 3.91 b | 0.95 a | 8.75 a | 3.82 b | 7.27 b | 5.19 b | 2.14 b | 2.50 a | 3.28 c | 0.45 b | 5.84 b | 3.35 b | 5.99 b | 4.63 c | 1.70 b | 1.90 b |
| N2 | | 3.78 b | 0.57 c | 6.84 b | 3.79 b | 6.95 b | 5.28 b | 1.79 c | 2.15 c | 3.66 b | 0.83 a | 8.44 a | 3.70 b | 7.15 a | 5.13 b | 2.44 a | 2.19 a |
| N3 | | 4.25 a | 0.73 b | 9.13 a | 4.51 a | 8.23 a | 6.95 a | 2.57 a | 2.08 c | 4.07 a | 0.81 a | 8.06 a | 4.36 a | 7.52 a | 6.06 a | 2.43 a | 2.29 a |
| | MW | 3.82 a | 0.79 a | 7.93 a | 3.87 a | 6.99 a | 5.35 b | 2.06 b | 2.41 a | 3.48 a | 0.59 a | 7.09 a | 3.68 a | 6.53 a | 5.10 a | 1.96 b | 1.99 a |
| | IW | 3.89 a | 0.77 a | 7.82 a | 3.94 a | 7.33 a | 5.64 a | 2.25 a | 2.12 b | 3.55 a | 0.62 a | 6.90 a | 3.69 a | 6.63 a | 5.03 a | 2.15 a | 2.06 a |
| Sig | | | | | | | | | | | | | | | | | |
| NL × PP | | *** | *** | *** | * | *** | *** | *** | *** | ns | *** | *** | ns | ns | ns | *** | *** |
| NL | | *** | *** | *** | *** | *** | *** | *** | *** | *** | *** | *** | *** | *** | *** | *** | *** |
| PP | | Ns | ns | ns | Ns | ns | * | *** | *** | ns | ns | ns | ns | ns | ns | *** | ns |

MW, mono-cropped wheat; IW, intercropping wheat. Values with different letters are significantly different among the N levels, planting pattern, and interaction of N levels and planting pattern according to Duncan's multiple range test (two-way ANOVA, $p < 0.05$). *, $p < 0.05$; ***, $p < 0.001$; ns, no significance.

**Table 5.** The fraction content of each non-essential amino acids in grain for inter- and mono-cropped wheat grain under different N levels.

| N Levels (NL) | Planting Patterns (PP) | 2019 | | | | | | | | | 2020 | | | | | | | | |
|---|---|---|---|---|---|---|---|---|---|---|---|---|---|---|---|---|---|---|---|
| | | Asp | Glu | Ser | Arg | Gly | Pro | Ala | Cys | Tyr | Asp | Glu | Ser | Arg | Gly | Pro | Ala | Cys | Tyr |
| N0 | MW | 5.81 ab | 26.30 de | 4.47 ef | 5.14 bc | 3.24a | 9.22 c | 5.33 abc | 1.07 a | 1.22 b | 4.03 d | 22.64d | 4.08 a | 4.27 e | 2.80 e | 9.81 a | 3.72c | 1.32a | 0.87 c |
| | IW | 4.87 cd | 26.56 cde | 4.50 ef | 4.59 ef | 3.23 a | 10.83 b | 5.39 ab | 1.09 a | 1.17 b | 4.67 bc | 24.09 cd | 4.44 a | 4.73 d | 3.07 d | 9.55 a | 4.16bc | 1.03b | 1.30 b |
| N1 | MW | 5.16 bcd | 21.98 e | 4.20 f | 4.59 c | 3.20 a | 9.34 c | 5.03 bc | 1.13 a | 1.10 c | 4.54 bc | 24.96 c | 4.38 a | 4.77 cd | 3.09 d | 10.27 a | 4.26 bc | 0.77 bc | 1.27 b |
| | IW | 6.03 ab | 32.97 b | 5.87 ab | 5.49 ab | 3.58 a | 11.35 b | 6.01 a | 0.47 c | 1.42 b | 4.45 c | 24.60 cd | 4.34 a | 4.92 cd | 3.24 cd | 10.31 a | 3.88 c | 1.42a | 1.28 b |
| N2 | MW | 6.15 a | 25.79 de | 5.11 cd | 5.13 bc | 3.61 a | 11.79 b | 5.78 a | 0.21 d | 1.36 b | 4.60 bc | 31.56 b | 4.97 a | 5.22 bc | 3.39 c | 11.43 a | 4.63 b | 0.98 b | 1.30 b |
| | IW | 4.36 d | 27.66 cd | 4.81 bc | 5.07 cd | 3.45 a | 11.16 b | 4.42 d | 0.70 b | 1.14 b | 6.27 a | 32.50 b | 4.92 a | 5.09 bcd | 3.49 bc | 11.72 a | 5.60 ab | 0.42 d | 1.20 b |
| N3 | MW | 5.53 abc | 38.36 a | 6.10 a | 5.82 a | 4.20 a | 16.50 a | 5.42 ab | 0.50 c | 1.76 a | 6.42 a | 38.62 a | 5.59 a | 6.19 a | 4.19 a | 14.40 a | 5.82 a | 0.76 bc | 1.52 a |
| | IW | 5.24 abcd | 31.40 bc | 5.45 bc | 5.63 bc | 3.93 a | 15.50 a | 4.67 cd | 0.59 bc | 1.43 b | 4.97 b | 32.23 b | 5.36a | 5.47 b | 3.72 b | 13.32 a | 4.64 b | 0.57cd | 1.46 a |
| N0 | | 5.34 a | 26.43 b | 4.48 c | 3.24 b | 3.24 c | 10.03 c | 5.36 a | 1.08 a | 1.20 b | 4.35 b | 23.36 d | 4.26 c | 4.50 d | 2.94 d | 9.68 c | 3.94 b | 1.17 a | 1.09 c |
| N1 | | 5.60 a | 27.48 b | 5.04 b | 3.20 b | 3.39 bc | 10.34 c | 5.52 a | 0.80 b | 1.26 b | 4.49 b | 24.78 c | 4.36 c | 4.84 c | 3.17 c | 10.29 c | 4.07 b | 1.09 a | 1.27 b |
| N2 | | 5.26 a | 26.73 b | 4.96 b | 3.61 b | 3.53 b | 11.48 b | 5.10 a | 0.45 c | 1.25 b | 5.43 a | 32.03 b | 4.95 b | 5.15 b | 3.44 b | 11.58 b | 5.12 a | 0.70 b | 1.25 b |
| N3 | | 5.39 a | 34.88 a | 5.77 a | 4.20 a | 4.06 a | 16.00 a | 5.05 a | 0.55 c | 1.60 a | 5.69 a | 35.42 a | 5.48 a | 5.83 a | 3.95 a | 13.86 a | 5.23 a | 0.66 b | 1.49 a |
| | MW | 5.66 a | 28.11 a | 4.97 a | 5.17 a | 3.56 a | 11.71 b | 5.39 a | 0.73 a | 1.36 a | 4.90 b | 29.45 a | 4.76 a | 5.11 a | 3.37 a | 11.48 a | 4.61 a | 0.96 a | 1.24 b |
| | IW | 5.13 b | 29.65 a | 5.16 a | 5.19 a | 3.55 a | 12.21 a | 5.12 a | 0.71 a | 1.29 a | 5.09 a | 28.36 b | 4.77 a | 5.05 a | 3.38 a | 11.23 a | 4.57 a | 0.86 a | 1.31 a |
| Sig | | | | | | | | | | | | | | | | | | | |
| NL × PP | | ** | *** | *** | ** | ns | *** | *** | *** | * | *** | *** | ns | *** | *** | ns | *** | *** | *** |
| NL | | Ns | *** | *** | *** | *** | *** | ns | *** | ** | *** | *** | *** | *** | *** | *** | *** | *** | *** |
| PP | | * | ns | ns | ns | ns | * | ns | Ns | ns | * | * | ns | ns | ns | ns | ns | ns | * |

MW, mono-cropped wheat; IW, intercropping wheat. Values with different letters are significantly different among the N levels, planting pattern, and interaction of N levels and planting pattern according to Duncan's multiple range test (two-way ANOVA, $p < 0.05$). *, $p < 0.05$; **, $p < 0.01$; ***, $p < 0.001$; ns, no significance.

## 4. Discussion

### 4.1. Effect of Cereal and Legume Intercropping on Grain Protein Content

GPC is an important index to reflect wheat quality; thus, it is of importance to simultaneously achieve high GPC and GY in wheat practice [35]. The present findings are in accordance with a previous study [17] that wheat and faba bean intercropping could simultaneously achieve both high GY and GPC because increased GPC in 2020 and increased protein yield in both years were found, and the intercropping effect was not influenced by N rates (Table 1). Yet, it was noted that GY and N uptake in intercropping depended on the maximum plant height, canopy, radiation use efficiency, interspecies interaction, the period of co-growing season, and so on [36]. Hence, conflict results of the effect of intercropping on GY and GPC were presented in different cereal-legume intercropping systems [22,36]. The wheat N uptake ability from flowering to maturity was one of the main reasons for the high GPC [37] and the N remobilization process was a potential target for improving the quality of wheat grain [20]. Recent studies found that wheat and faba bean intercropping stimulated wheat N uptake during mid- and late- growth stages and induced more N to shift from straws to grain due to intercropping up-regulating the key N assimilation enzyme activity and gene expression during the reproductive growth stages [38,39]. Thus, it could partly explain the reason for intercropping increasing GPC in the present study. Some temporary legume-based intercropping patterns were adopted in many regions due to overcoming some problems including technical and competition in intercropping, and in such conditions, legumes usually improved soil N availability for cereal and finally changed cereal GY and GPC [23,40]. In the present study, we observed that continuous intercropping increased soil N availability especially under low-N-input conditions (Supplementary Table); thus, we could not distinguish the role of the long- and short-term intercropping in improved GPC and GY.

In the present study, increased gliadins in both years and increased glutenins in 2019 were found due to wheat intercropped with faba bean (Figure 3). Gliadins and glutenins content determined the bread-making characteristics of wheat [41], because they play an important role in dough rheology [42]. These results in the present study meant that intercropping could alter the end-use of wheat quality, and more studies are needed to elucidate the mechanism of intercropping modulating protein fractions and their role related to wheat grain quality.

The present finding is partly in accordance with the results of a global meta-analysis that split N had a greater effect on wheat yield and protein content in less fertile soils and at high N rates [43], because GPC was increased by N input in 2020 but was not influenced by N rate in 2019 at the current situation (Table 1). Thus, N management is still a good strategy to improve GPC in the southwest of China. In a previous study, we found that wheat and faba bean had potential to save N input but still maintain wheat grain yield [21]; however, according to the present study, we could not ascertain whether decreased N input in intercropping would affect wheat GPC. This suggests that both GY and GPC should be taken into account when establishing an optimal N rate in the cereal and legume intercropping system.

### 4.2. Effect of Cereal and Legume Intercropping on Grain Amino Acids Content

In the present study, we found that the effect of intercropping on AAs content including NEAAs and EAAs was dependent on N levels, because some EAAs and NEAAs fractions declined due to intercropping when N was overused (N3 level), but some AAs fractions increased when wheat was intercropped with faba bean at low N levels (N0 and N1 level) (Tables 4 and 5). Taken together, the effect of intercropping on GPC was not affected by N rates in the present study, but it seems that wheat N input should not exceed 180 kg ha$^{-1}$ in intercropping, because intercropping declined wheat AAs content at the N3 level (Table 1). Actually, wheat protein quality is not only dependent on the protein content but also related to the balance of AAs [44]. However, few studies have focused on intercropping on cereal

AAs content. Thus, the findings in the present study suggest that modulating N rates should be imperative to wheat grain quality in the legume-based intercropping system.

High NEAAs, especially high Pro and Glu content, were found in the present study (Table 5), which is in accordance with a previous study [45], whereas NEAAs such as Pro and Glu have a low nutritional value; thus, improvement in EAAs is more important for wheat grain quality. In the present study, it seems that intercropping did not modulate the ratio of EAAs to NEAAs, though there was year's variation (Figure 4), and intercropping had a greater impact on wheat grain protein rather than AAs. These findings should be linked with N remobilization and protein production during grain development. Still, more work on AAs and protein synthesis in intercropping could fully understand the findings.

### 4.3. Cereal and Legume Intercropping Modulated the Relationship between Grain Yield and Quality

The present study supports a previous study that when agronomic practices were given consideration, there was no trade-off between GY and quality [46], because we found steady GPC (10–15%) with increasing GY for MW, and GPC tended to increase with increasing GY for IW (Figure 5). Actually, wheat GPC content was largely dependent on post-anthesis N uptake [26]. Hence, the shift in the enhanced wheat N from the leaves and the stem to the grain and the stimulated wheat growth rate during the wheat mid-growing season [39,47] should be responsible for the changed correlation between GY and GPC in intercropping. The rainfall and temperature during 2018/2019 and 2019/2020 growing seasons were different (Figure 1), which might induce the effect of intercropping and N levels on GPC, which was different year by year in the present study (Table 1). However, the intercropping yield advantage was stable in the two-year field experiment (Table 1); thus, we thought that grain quality might be more sensitive to temperature and rainfall than grain yield. Hence, no relationship between GY and GPC was found for MW in the present study, but more work should still be conducted in the future to ascertain the correlation between GY and GPC under the current situation.

An early study from Eppendorfer found that correlations between AAs and N content within a variety were similar [48]. However, the correlation between wheat, maize, and soybean GPC and AAs presented a high variation [45,49]. In the present study, linear regression equations were established and significant correlations were found both between AAs and GCP and between AAs and GY for mono- and inter-cropped wheat grain (Figure 5), but we did not analyze the relationship between each AA and GPC and GY. According to our findings, intercropping either increased or decreased some specific AAs content, and intercropping affected the contents of TAAs, EAAs, and NEAAs in wheat grain under different N levels; hence, it could deduce the relationship between GPC and the given AA, which should change due to intercropping.

### 5. Conclusions

Higher protein yield and AAs yield were obtained when wheat was intercropped with faba bean. Intercropping mainly increased wheat GPC rather than AAs content because intercropping had no impact on AAs content regardless of N levels, but the 9% GPC of IW was higher than that of MW in 2020. Wheat gliadin content was increased on average by 8–14% when intercropped with faba bean. Similarly, some EAAs and NEAAs fraction contents were increased due to intercropping under N0 and N1 levels, but IW presented lower contents of EAAs and NEAAs fractions at the N3 level relative to MW. There was no trade-off relationship between GPC and GY according to regression analysis in the present study, and intercropping was a good option for simultaneously achieving both high GY and GPC. Hence, wheat and faba bean intercropping presented a potential to improve both wheat grain quality and yield, and modulated N rates were important to maximize the intercropping advantage in terms of grain quality. We suggest that the wheat N application rate should not exceed 180 kg ha$^{-1}$ to achieve both intercropping yield and quality advantages in the southwest of China and similar regions.

**Supplementary Materials:** The following supporting information can be downloaded at: https://www.mdpi.com/article/10.3390/agronomy12122984/s1, Table S1: Soil total nitrogen and available nitrogen contents in the each treatment at soil depths of 0–20cm before the start of the experiment of 2018–2019.

**Author Contributions:** Conceptualization, J.X. and Y.Z.; methodology, Y.D. and L.T.; formal analysis, Y.-a.Z., J.H. and Z.Y.; investigation, Y.-a.Z., J.H., Z.Y., D.Z., H.L., X.W.; writing—original draft preparation, Y.-a.Z.; writing—review and editing, J.H.; funding acquisition, J.X. All authors have read and agreed to the published version of the manuscript.

**Funding:** This work was supported by the National Natural Science Foundation of China (32060718 and 31760611) and the Yunnan Agricultural Foundation Joint Project (2018FG001-071).

**Data Availability Statement:** Not applicable.

**Acknowledgments:** We thank LetPub (www.letpub.com) for its linguistic assistance during the preparation of this manuscript.

**Conflicts of Interest:** The authors declare no conflict of interest.

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
