# Peer review of "Wheat and Faba Bean Intercropping Together with Nitrogen Modulation Is a Good Option for Balancing the Trade-Off Relationship between Grain Yield and Quality in the Southwest of China"

_agronomy, doi:10.3390/agronomy12122984_

Round 1
Reviewer 1 Report
This manuscript titled "Wheat faba bean intercropping is a good option for balance the trade off relationship between grain yield and quality in southwest China" assess intercropping impact on relationship between crop yield and quality. The research study will provide unique contribution of intercropping technology on grain quality with specific to protein and associated amino acids which are essential for human health. However, I suggest the following to improve the quality of the manuscript:
1. Abstract:
1.1 The purpose of the study is not clear in the Abstract and author(s) need to state the purpose clearly.
1.2 Sentence from Line 33 to Line 37 is too long and needs to be re-written whereby the Nitrogen Levels must be moved to form part of Materials and Methods. The sentence will then read as follows: :Two planting patterns which are mono-cropped wheat (MW), intercropped wheat + faba bean (IW) and four nitrogen levels was designed with three replications".
2. Introduction
2.1 The hypothesis must be formulated clearly.
2.2 At the end of the Introduction, author(s) must indicate how the paper is structured.
3. Materials and Methods
3.1 Field experiment site:
-It is important for author(s) to indicate clearly right on the onset when the current study was conducted. At present it is confusing about when exactly the current study was conduct as sometimes the author(s) refer to 2014 as the year the study was established and in other parts they refer to 2018/2019. To avoid this confusion, the author(s) need to have sentence formulated as follows: "The grain yield, protein content and amino acids were studied during 2018/2019 and 2019/2020 cropping seasons in the existing Wheat-Faba bean Intercropping Experiment which was established in 2014".
-In Line 108, indicate whether the field experiment was conducted at the Research Site or at the farmer's land.
-The location of the field experiment, climatic conditions of the study area, dominated soil type indicated with key soil properties like texture, depth, and soil name according to Local soil classification system and FAO soil classification system, etc. must form part of the characteristics of Experiment Site.
-Detailed information of field experiment site must be provided in the current study instead of referring readers to previous study (see Line 108 to Line 109). This information is important for replicability of the study in other parts of the world for the benefit of the readers.
3.2 Experimental design
-Sentences from Line 132 to Line 139 must be moved to Field experiment management as these are not design but cropping management.
3.3 Sample collection and measurements
-The first sentence in Line 149 to Line 151 is confusing as it refers to 2014. Concentrate on 2018/2019 and 2019/2020 experiment activities. Indicate whether harvesting was done in the whole plot or on selected rows. Rephrase this sentence.
-Regression equations in Line 179 to Line 190 are currently clustered and this must be presented properly. Refer to other reference studies how these are presented.
3.4 Data collection
-This part is not clear. Also one sentence cannot be a paragraph. I suggest you remove this to Sample collection and measurement section. 2.4 subsection will now be: "Sampling, data collection and measurement"
3.5 Statistical analysis
-Provide version of SPSS, e.g. IBM SPSS Statistics Version 23
4. Conclusions
-In this section I suggest that you also provide limitations of your study and also propose future research.
5. Tables
-These need to be properly formatted in accordance with MDPI standards.
-For each footnote of the Tables at the bottom, I suggest you add "according to Duncan's multiple range test" at the end of each footnote. This will indicate that your statistical analysis is in accordance with what is indicated in Materials and Methods under section Statistical analysis.
-In Table 3 you also need to indicate the meaning of the following symbols *; **; *** as the footnote at the bottom of the Table as you did in other Tables
6. Contribution by each author must be provided
7. Funding of the study must be indicated.
Author Response
Point by Point Responses to the Issues Raised by the Reviewers for agronomy-1992558
Reviewer #1: Comments:
This manuscript titled "Wheat faba bean intercropping is a good option for balance the trade off relationship between grain yield and quality in southwest China" assess intercropping impact on relationship between crop yield and quality. The research study will provide unique contribution of intercropping technology on grain quality with specific to protein and associated amino acids which are essential for human health. However, I suggest the following to improve the quality of the manuscript:
- Abstract:
1.1 The purpose of the study is not clear in the Abstract and author(s) need to state the purpose clearly.
*** Thank you so much! We have carefully revised according to the suggestions.
1.2 Sentence from Line 33 to Line 37 is too long and needs to be re-written whereby the Nitrogen Levels must be moved to form part of Materials and Methods. The sentence will then read as follows: :Two planting patterns which are mono-cropped wheat (MW), intercropped wheat + faba bean (IW) and four nitrogen levels was designed with three replications".
*** Good suggestion! We have done according to the suggestions.
- Introduction
2.1 The hypothesis must be formulated clearly.
*** Thank you so much! The hypothesis has been clearly described in the revised manuscript.
2.2 At the end of the Introduction, author(s) must indicate how the paper is structured.
*** Good suggestion! We have done according to the suggestions.
- Materials and Methods
3.1 Field experiment site:
-It is important for author(s) to indicate clearly right on the onset when the current study was conducted. At present it is confusing about when exactly the current study was conduct as sometimes the author(s) refer to 2014 as the year the study was established and in other parts they refer to 2018/2019. To avoid this confusion, the author(s) need to have sentence formulated as follows: "The grain yield, protein content and amino acids were studied during 2018/2019 and 2019/2020 cropping seasons in the existing Wheat-Faba bean Intercropping Experiment which was established in 2014".
We carefully revised the manuscript according to the suggestion, and we think the revised manuscript should be clear.
-In Line 108, indicate whether the field experiment was conducted at the Research Site or at the farmer's land.
*** Thank you so much! The field experiment was conducted at the Research Site, we have clearly stated it in the revised manuscript.
-The location of the field experiment, climatic conditions of the study area, dominated soil type indicated with key soil properties like texture, depth, and soil name according to Local soil classification system and FAO soil classification system, etc. must form part of the characteristics of Experiment Site.
*** Thank you so much! All these information including soil texture, soil type, and climate were clearly stated in the manuscript.
-Detailed information of field experiment site must be provided in the current study instead of referring readers to previous study (see Line 108 to Line 109). This information is important for replicability of the study in other parts of the world for the benefit of the readers.
*** Good suggestion! We have provided these information in the revised manuscript.
3.2 Experimental design
-Sentences from Line 132 to Line 139 must be moved to Field experiment management as these are not design but cropping management.
*** Good suggestion! We have revised according to the suggestion.
3.3 Sample collection and measurements
-The first sentence in Line 149 to Line 151 is confusing as it refers to 2014. Concentrate on 2018/2019 and 2019/2020 experiment activities. Indicate whether harvesting was done in the whole plot or on selected rows. Rephrase this sentence.
*** Good suggestion! We have rephrased this sentence, and it is clear now in the revised manuscript.
-Regression equations in Line 179 to Line 190 are currently clustered and this must be presented properly. Refer to other reference studies how these are presented.
*** Good suggestion! We revised this part according to other reference studies.
3.4 Data collection
-This part is not clear. Also one sentence cannot be a paragraph. I suggest you remove this to Sample collection and measurement section. 2.4 subsection will now be: "Sampling, data collection and measurement".
*** Good suggestion! We have revised according to the suggestions.
3.5 Statistical analysis
-Provide version of SPSS, e.g. IBM SPSS Statistics Version 23
*** Thank you so much! We have revised according to the suggestions.
- Conclusions
-In this section I suggest that you also provide limitations of your study and also propose future research.
*** Thank you so much! We have revised according to the suggestions.
- Tables
-These need to be properly formatted in accordance with MDPI standards.
-For each footnote of the Tables at the bottom, I suggest you add "according to Duncan's multiple range test" at the end of each footnote. This will indicate that your statistical analysis is in accordance with what is indicated in Materials and Methods under section Statistical analysis.
-In Table 3 you also need to indicate the meaning of the following symbols *; **; *** as the footnote at the bottom of the Table as you did in other Tables
*** Thank you so much! We have revised according to the suggestions.
- Contribution by each author must be provided
*** Thank you so much! We provided the contribution of each author in the revised manuscript.
- Funding of the study must be indicated.
*** Thank you so much! We provided the funding in the original manuscript, and we also carefully checked in the revised manuscript.
Reviewer 2 Report
Interesting research results for science and agricultural practice. I appreciate that this is multi-year and field research. I have included my comments in the text of the manuscripts. The entire text should be prepared in accordance with the requirements of the Agronomy journal. Why were the yields of wheat not shown in t / ha? Add numbering for all chapters. Write a research hypothesis. The chapter "Methodology and Methods" requires the most corrections. Notes in the text of the manuscript. Briefly describe the weather conditions during the years of research. Check the numbers of figures and tables and cross-references in the text of the manuscripts. In conclusion, add which nitrogen dose is recommended for agricultural practice. Correct the literature list as required by the journal. Enter the Latin names of the species in italics. After the improvement, I recommend publishing the manuscript in the journal Agronomy.

Author Response
Point by Point Responses to the Issues Raised by the Reviewers for agronomy-1992558
Thank you so much! We have accepted almost all the suggestions made by reviewer #2.
The reviewer pointed out that a conversion factors of 5.83 was not reliable. But, we think it should be correct. In the present study, grain protein content (GPC) was estimated by multiplying grain N concentration with a conversion factors of 5.83 for wheat, which was according to FAO 2003. In addition, this conversion factor was adopted by many researchers and published in many peer reviewed paper. For example:
Wang, J.; Hasegawa, T.; Li, L.; Lam, S K.; Zhang, X.; Liu, X.; Pan, G. Changes in grain protein and amino acids composition of wheat and rice under short term increased [CO2] and temperature of canopy air in a paddy from East China. New Phytologist 2019, 222:726–7
Berecz, K.; Simon‐Sarkadi, L.; Ragasits, I.; Hoffmann, S. Comparison of protein quality and mineral element concentrations in grain of spelt (Triticum spelta. L.) and common wheat (Triticum aestivum L.). Arch Agron Soil Sci, 2001, 47, 389–398.
Thus, we think it should be reliable.

Reviewer 3 Report
The manuscript with the title “Wheat and faba bean intercropping is a good option for balance the tradeoff relationship between grain yield and quality in southwest China” presents the results of a intercropping experiment with effect on wheat assessed over a few years.
Abstract
is excessively detailed and long. Experimental factors should be presented concise here. Results mentioned in the abstract should highlight only the most interesting/novel findings. For example “a two-year bi-factorial trial was conducted, to investigate the role of two planting patterns (…) and four nitrogen fertilization levels (…), as well as their interaction on productivity of wheat (yield and grain quality). Results showed that … ”
Please be careful at spelling
Line 79 correct to - planting
Lines 85, 87 correct to - durum wheat
Material and Method section
Experiment took place between 2014-2020, please check in the abstract and results section how many agronomic crop years/harvest seasons were specified as studied (in the abstract/some results chapters are mentioned two).
Line 109
Refers here that previous crop was maize (corn)? The most recent rotation history of the field is not expressed sufficiently clear, corn mono-culture was the previous crop?
Line 149
If the crop was established in 2014, the first harvest was in 2015, and no harvest in 2014, right? …
Please clarify the harvest years, and experimental years intervals.
Statistical analysis
Please clearly state the influence of which factors and interactions were examined and why. No climatic data is mentioned in the material and method section.
I see that only the influence of the two factors were analyzed, but not the interaction years x treatments. Why were these considered not relevant here, does it have to do with the local climate?
Results
I am not certain it is the best way to say that a specific compound/content was “regulated” by N level and/or planting pattern. I think the agronomically correct way is to say that specific compounds from grain were “influenced” or that N level “exercised an influence”, or alternatively that there was identified a relationship between…., or that genotype potential was maximized under a specific N level or treatment. Because “regulation” in this context does not seem to me the best choice of words. Please use more agronomic terms and expressions.
Results 3.1.
Interesting that bi-partite interaction N level x planting pattern did not exercise influence on grain protein content. However, Lines 302-305 from Discussion support the idea that a benefit was identified regarding treatments. … Please find a coherent way to highlight important findings in relation to the expressed objectives and insist on those, otherwise the manuscript becomes overloaded and statements become overwhelming to the point of confusion for the reader.
Moreover, N level alone did not influence grain protein content in 2019. Considering the fact that there were differences between years, it’s a pity that influence of N treatments x years interaction was not considered, as it appears it might have been relevant. What was different in 2019 about the local climate that could have influence the crop and the results compared to 2020?
Statements Line 214-215 and Line 2019-2020 seem contradictory or confusing: did or did not N level influence that GPC?
Line 293 “no relationship between grain yield (GY) and grain protein content (GPC) was found for mono-cropped wheat (MW)”. Seems unusual, as in wheat generally the yield and grain protein concentration are usually negatively correlated, and an effect/trend should be seen. In addition, Nitrogen (N) nutrition is the major management factor agronomists use to alter grain protein. Here in this study the doses were different enough, and one would expect some differences and patterns to emerge.
Unfortunately, there is no clear pattern emerging following the results and perhaps it has to do with the presentation style, since it is a bit confusing. The text would benefit from increase clarity and editing for better readability.
Discussion
I advise authors to underline main findings that could have practical implications, and maintain coherence across the manuscript in accordance with the objectives defined first.
Best regards.
Author Response
Point by Point Responses to the Issues Raised by the Reviewers for agronomy-1992558
Reviewer #3: Comments:
The manuscript with the title “Wheat and faba bean intercropping is a good option for balance the tradeoff relationship between grain yield and quality in southwest China” presents the results of a intercropping experiment with effect on wheat assessed over a few years.
Abstract
is excessively detailed and long. Experimental factors should be presented concise here. Results mentioned in the abstract should highlight only the most interesting/novel findings. For example “a two-year bi-factorial trial was conducted, to investigate the role of two planting patterns (…) and four nitrogen fertilization levels (…), as well as their interaction on productivity of wheat (yield and grain quality). Results showed that … ”
*** Good suggestion! We have carefully revised according to the suggestions.
Please be careful at spelling
Line 79 correct to - planting
Lines 85, 87 correct to - durum wheat
*** Thank you so much! Indeed, we make mistakes, and we have carefully checked the entire manuscript.
Material and Method section
Experiment took place between 2014-2020, please check in the abstract and results section how many agronomic crop years/harvest seasons were specified as studied (in the abstract/some results chapters are mentioned two).
*** Thank you so much! The present study was based on a multi-year field experiment established in 2014, but we only used the data collection in the 2018/2019 and 2019/2020 growing seasons. We have clearly stated this point, and we think it should be clear now.
Line 109
Refers here that previous crop was maize (corn)? The most recent rotation history of the field is not expressed sufficiently clear, corn mono-culture was the previous crop?
*** Thank you so much! The previous crop was corn, and corn mono-culture was used for many years before established the present field experiment. This statement has been provided in the revised manuscript.
Line 149
If the crop was established in 2014, the first harvest was in 2015, and no harvest in 2014, right? …
Please clarify the harvest years, and experimental years intervals.
*** Thank you so much! The present study was based on a multi-year field experiment established in 2014, but we only used the data collection in the 2018/2019 and 2019/2020 growing seasons. We have clearly stated this point, and we think it should be clear now.
Statistical analysis
Please clearly state the influence of which factors and interactions were examined and why. No climatic data is mentioned in the material and method section.
I see that only the influence of the two factors were analyzed, but not the interaction years x treatments. Why were these considered not relevant here, does it have to do with the local climate?
*** Thank you so much! The climatic data was supplied in the revised manuscript, and the temperature and rainfall were different during two growing seasons. Indeed, we did not conducted the interaction years x treatments due to climate difference. We have discussed this point in the section of discussion in the revised manuscript.
Results
I am not certain it is the best way to say that a specific compound/content was “regulated” by N level and/or planting pattern. I think the agronomically correct way is to say that specific compounds from grain were “influenced” or that N level “exercised an influence”, or alternatively that there was identified a relationship between…., or that genotype potential was maximized under a specific N level or treatment. Because “regulation” in this context does not seem to me the best choice of words. Please use more agronomic terms and expressions.
*** Good suggestion! We have carefully revised according to the suggestions.
Results 3.1.
Interesting that bi-partite interaction N level x planting pattern did not exercise influence on grain protein content. However, Lines 302-305 from Discussion support the idea that a benefit was identified regarding treatments. … Please find a coherent way to highlight important findings in relation to the expressed objectives and insist on those, otherwise the manuscript becomes overloaded and statements become overwhelming to the point of confusion for the reader.
*** Thank you so much! We have clearly revised, and it should be clear now.
Moreover, N level alone did not influence grain protein content in 2019. Considering the fact that there were differences between years, it’s a pity that influence of N treatments x years interaction was not considered, as it appears it might have been relevant. What was different in 2019 about the local climate that could have influence the crop and the results compared to 2020?
*** Thank you so much! Indeed, the temperature and rainfall were different during two growing seasons. We have discussed this point in the section of discussion in the revised manuscript.
Statements Line 214-215 and Line 2019-2020 seem contradictory or confusing: did or did not N level influence that GPC?
*** Thank you so much! N level had no impaction on GPC in 2018/2019. We have clearly stated it in the revised manuscript.
Line 293 “no relationship between grain yield (GY) and grain protein content (GPC) was found for mono-cropped wheat (MW)”. Seems unusual, as in wheat generally the yield and grain protein concentration are usually negatively correlated, and an effect/trend should be seen. In addition, Nitrogen (N) nutrition is the major management factor agronomists use to alter grain protein. Here in this study the doses were different enough, and one would expect some differences and patterns to emerge.
Unfortunately, there is no clear pattern emerging following the results and perhaps it has to do with the presentation style, since it is a bit confusing. The text would benefit from increase clarity and editing for better readability.
*** Thank you so much! Usually, a negative relationship was found between the wheat grain yield (GY) and grain protein content, but in the present study no relationship between GY and GPC was found for mono-cropped wheat. The present study might be related to N application rates had no impaction on GPC in 2018/2019 growing season. We have discussed it in the section of discussion. Actually, we still could not determine the relationship between GY and GPC based on current results. Thus, we think more data collected from different years should be important to ascertain relationship between GY and GPC. This point we have clearly stated in the discussion section in the revised manuscript.
Discussion
I advise authors to underline main findings that could have practical implications, and maintain coherence across the manuscript in accordance with the objectives defined first.
*** Good suggestion! We have carefully revised according to the suggestions.

Round 2
Reviewer 3 Report
Dear authors,
thank you for your comprehensive changes and improvements to the manuscript. Although I have some reservations regarding some claims and results, these were the results you obtained after all and therefore I have no reason to oppose the publication.
Best regards.
Author Response
Encl: Point by Point Responses to the Issues Raised by the Reviewers for agronomy-1992558 -R1
Line 120. “The wheat and faba bean intercropping experiment established in 2014, was 120 conducted….” : The reference to year of experiment again is a redundancy since it is already mentioned in the previous sentence.
***Thank you! We have revised as suggestion.
Line 124-125: the average minimum and maximum temperatures is 14°C. (This is not clear?)
***Thank you! We have rewrite the sententce in the re-revised manuscript.
Line 128: “The soil was a cultivated red soil…. “ : This is not a proper scientific English writing…” cultivated red soil “ Here you could include soil classification.
*** Good suggestion! The soil in this study was a red soil. We have clearly stated in the re-revised manuscript.
In Lines 122-125 the authors are referred to climate and then again at the end of this section they are also referred to the meteorological data. Please put these together somehow.
**** Good suggestion! We have revised according to suggestion.
Subsection 2.1. Title could be: “Experimental site and growing conditions”
*****Thank you so much. We have revised as suggestion.
Lines 149-151. Writing concerning spacing of plots is not clear. Please rewrite clearly. Suggestion: N the present study wheat row spacing was 0.2 m with seeding rate at 180 kg ha-1, and plant-to-plant spacing of 0.1 m whereas, faba bean row spacing was 0.3-m.
*****Thank you so much. We have revised as suggestion. But it is not good to describe the wheat plant space, because there is tillering stage for wheat, and it is diffcult to determine plant space.
Line 167: the local varieties….were…..
**** Thank you so much. We have carefully revised the entire manuscript.
Subsection 2.4. Title could be “Data collection and analyses”. Also the authors present in their results Grain yield and grain protein yield. Therefore they should mention that: “At physiological maturity on xx 2018/2019 and on xx 2019/2020 growing season grain yield of inter- and mono- cropped winter wheat was determined from an area of (or on whole plot basis?) ….after drying the seeds at 65 C for at least 48 h.???? …..”
****Good suggestion. We have revised them as suggestion. Actually, the peer reviewer #1 suggested that using Sampling, data collection and measurement in the revised manuscript. Taken together, we think Data collection and analyse should be better. Thanks again.
Also authors present data on grain protein yield. Therefore they should somewhere mention that “grain protein yield is the product of grain yield and grain protein content”.
****Thank you! We have clearly stated this point in the orginal manuscript, at the end of section 2.4
Line 252: was neither affected by N levels………
****Thank you! We have revised, and we think it should be clear.
The whole manuscript should be corrected for the English scientific writing.
*****Thank you so much. We have carefully revised the entire manuscript.